# Factors associated with symptoms of major depression disorder among transgender women in Northeast Brazil

Marcelo Machado de Almeida[1]◉*, Luís Augusto Vasconcelos da Silva[1,2‡], Francisco Inácio Bastos[3]◉, Mark Drew Crosland Guimarães[4]◉, Carolina Coutinho[5‡], Ana Maria de Brito[6‡], Socorro Cavalcante[7‡], Inês Dourado[1]◉

1 Institute of Collective Health, Federal University of Bahia, Salvador, Brazil, 2 Institute of Arts, Sciences and Humanities, Federal University of Bahia, Salvador, Brazil, 3 Oswaldo Cruz Foundation, Rio de Janeiro, RJ, Brazil, 4 Department of Preventive and Social Medicine, Faculty of Medicine, Federal University of Minas Gerais, Belo Horizonte, Brazil, 5 Getúlio Vargas Foundation, São Paulo/SP, Brazil, 6 Aggeu Magalhães Institute, Oswaldo Cruz Foundation, Recife, Pernambuco, Brazil, 7 Health Department of the State of Ceará and Municipality of Fortaleza, Fortaleza, Brazil

◉ These authors contributed equally to this work.
‡ LAVS, CC, AMB and SC also contributed equally to this work.
* marmachadoalmeida@gmail.com

**Data Availability Statement:** All relevant data are within the paper and its Supporting Information files.

## Abstract

### Introduction

Transgender women (TGW) are one of the most vulnerable groups, including higher prevalence of HIV and mental health disorders, such as anxiety and depression than in the general population. Major Depression Disorder (MDD) is one of the most important mental health conditions due to an increasing trend in prevalence in the general population. This study aims at describing the prevalence of symptoms of MDD (SMDD) and associated factors among TGW in capitals of three States in Northeast Brazil.

### Methods

TGW n = (864) were selected from the cities of Salvador (n = 166), Recife (n = 350), and Fortaleza (n = 348) using Respondent Driven Sampling methodology. Symptoms of MDD were defined according to the Patient Health Questionnaire-9 scale. Multinomial logistic regression was used to compare those with mild/moderate or moderately severe/severe symptoms of depression with those with no depression, respectively, using complex sample design. Weighted Odds Ratio with 95% confidence interval were estimated.

### Results

51.1% of the sample was classified as mild/moderate and 18.9% as moderately severe/severe SMDD. Mild/moderate SMDD was associated with a history of sexual violence (OR = 2.06, 95%CI: 1.15–3.68), history of physical violence (OR = 2.09, 95%CI: 1.20–3.67),) and poor self-rated quality of life (OR = 2.14, 95%CI: 1.31–3.49).). Moderately severe/severe SMDD was associated with history of sexual violence (OR = 3.02, 95%CI: 1.17–

**Funding:** This study was supported by Brazilian Ministry of Health, through its Secretariat for Health Surveillance and its Department of Prevention, Surveillance and Control of Sexually Transmitted Infections, HIV/AIDS and Viral Hepatitis in the form of a grant (Project 914BRZ1138 BRAZIL AIDS-SUS) to FIB, on behalf of the DIVAS research group. The study was also supported by Fundação de Amparo à Pesquisa do Estado do Rio de Janeiro (FAPERJ) in the form of a grant (E-26/010.002428/2019) to FIB. The funders had no role in study design, data collection and analysis, decision to publish, or preparation of the manuscript.

**Competing interests:** The authors have declared that no competing interests exist.

7.77), history of physical violence (OR = 4.34, 95% CI:1.88–6.96), poor self-rated quality of life (OR = 3.32, 95%CI:1.804–6.12), lack of current social support (OR = 2.53, 95%IC: 1.31–4.88) and lack of family support in childhood (OR = 2.17, 95%IC 1.16–4.05)).

## Conclusions

Our findings strengthens the evidence of a higher prevalence of SMDD among TGW as compared to the general population. Public health policies and actions that target social determinants of risk and protection for MDD among TGW must be urgently implemented.

## Introduction

Persons who identify themselves under non-conforming gender performances have greater odds of developing various health problems when compared with the general population. Among them, transgender women (TGW) are one of the most vulnerable groups [1]. Specifically, mental disorders such as anxiety and mood disorders are more prevalent in TGW than in the general population [2, 3].

Major Depression Disorder (MDD) is one of the most important mental health conditions due to an increasing trend in prevalence in the general population, and it has been estimated by the WHO as the putative first cause of disability in 2030 [4]. In addition to disability, MDD is also associated with a higher risk of somatic diseases and other mental disorders, besides being the leading cause of attempted or committed suicide [5]. It should also be considered that it is strongly associated with the abuse of psychoactive substances and exposure to risky behaviors, such as unprotected sexual practices and greater exposure to episodes of sexual, physical, and psychological violence [6–10].

Studies estimate that the prevalence of MDD during one´s lifetime varies from 6% to 17% in the general population [11]. Other studies indicate higher prevalence among women (19.7%) [5, 8], and even higher rates among TGW as indicated below. Psychosocial factors such as experiences of violence and discrimination and other situations of social vulnerability, such as low levels of schooling, poverty, commercial sex work, lack of social support, living alone, poor perception of life quality, poor housing conditions, unemployment, use/abuse of psychoactive substances, and judicial problems are associated with MDD in TGW [7, 12–17]. Furthermore, TGW sex workers are especially vulnerable and more prone to MDD [14, 18, 19]. Bockting et al. (2013) found a positive association between stigma and MDD, as well as a negative association between transgender pride and psychological distress with MDD [15]. Nemoto et al. (2011) suggested that the perception of social support might be a more important protective factor than social support itself [14]. Family support during childhood as well as during transition seems to be an important protective factor for MDD in TGW [20, 21]. However, the families seldom accept TGW.

The association between living with HIV/AIDS and MDD is also well studied, and one of the factors that interfere in the causal path of this association is the use of antiretroviral therapy (ART), which may have important interactions with drugs used to treat MDD, depending on the drug regimens [22, 23]. Moreover, TGW have less access to health services in general, due to discrimination based on gender identity, social interactions and life trajectories [24]. A multicenter study conducted in European countries reported 60.0% transgender individuals have had at least one episode of affective disorders in their lifetime [25]. A study conducted in Ivory Coast with TGW and men who have sex with men (MSM), found a prevalence of MDD of

22.7% and 12.2%, respectively [26]. A study conducted in Italy, estimated a 42% prevalence of MDD among transgender people [27]. Finally, studies conducted in Italy and Brazil estimated the prevalence of MDD in 42.0% among transgender people and in 80.5% among TGW, respectively [27, 28]. Nevertheless, data on MDD in this population are still scarce, mainly when considering the continental dimensions, as well as the social, cultural, economic and political heterogeneities in Brazil. Therefore, this study aims to describe the prevalence of SMDD and associated factors among TGW in the Northeast Brazil.

## Materials and methods

This is an analysis of TGW data collected in the three largest cities in Northeast Brazil (Salvador, Fortaleza and Recife) that composed the DIVAS study (National Research Study on Behaviors, Attitudes, Practices and Prevalence of HIV, Syphilis and Hepatitis B and C among *Travestis* and Transsexual Women), a survey conducted in 12 cities in Brazil from November 2016 to June 2017, aimed at estimating the prevalence of HIV, and other sexually transmitted infections (STIs) and monitoring risk practices for these infections [29].

TGW (n = 864) were selected from the cities of Salvador (n = 166), Recife (n = 350) and Fortaleza (n = 348). Participants were recruited using Respondent Driven Sampling (RDS) methodology, in which participants themselves recruit their acquaintances, using a coupon system [29, 30]. A maximum of three recruitees was allowed per participant in order to reduce recruitment homophily. The eligibility criteria were: to be 18 years old or over; to identify herself as a *travesti* (emic concept to describe a specific gender identity in Brazil), woman, trans woman, or other female gender identification; to have been registered as male at birth; to spend most of the day in the studied municipalities. They also had to present a valid invitation coupon, agree to participate in the study and sign the informed consent form. TGW who were under the effect/influence of drugs and alcohol, during the interview, in a manner that rendered it difficult for them to understand the research questions, were excluded.

First participants ("seeds") were selected, after qualitative formative research, in an attempt to better assess the heterogeneity of the TGW population, according to demographic and socioeconomic conditions. In each city, 5–10 seeds launched the recruitment process. Each seed, and later each participant received three coupons to invite another TGW from their social contact network (referral chains). For a successful recruitment, RDS includes primary and secondary incentives. The primary one was U$ 10.00 as a compensation for transportation and lost worktime. The secondary one as a compensation for the recruitment of contacts was U$ 10.00 for each TGW recruited for the study. Data were collected through interviews with a standardized pre-tested questionnaire, conducted face-to-face by interviewers duly trained, in a space reserved exclusively for this purpose.

### Definition of SMDD and study variables

SMDD was defined according to the *Patient Health Questionnaire-9* (PHQ-9) scale, based on the diagnostic criteria of the *Diagnostic and Statistical Manual of Mental Disorders-Fourth Edition* for MDD, and it has been validated in Brazil [31]. This scale is indicated for screening of MDD in primary health care settings. For this analysis, SMDD were categorized into three groups: absence (score <5 points), mild/moderate (score 5–14 points) and moderately severe/severe depressive symptoms (score 15–27 points). Other variables used in this study were: 1- socio-demographics (age, schooling, monthly income, race/skin color, religion, and sex work over lifetime); 2- lifetime history of violence and discrimination (episodes of sexual, psychological and physical violence, history of police arrest, and experience of discrimination); 3- social and family support (current social support, transgender pride; family support in childhood,

family support during gender transition, self-rated quality of life, living alone at the time of the interview); and 4- health-related issues at the time of interview (use of hormones, access to health services, living with HIV/AIDS, use of ART).

Lifetime histories of having suffered physical, psychological or sexual violence were based on specific questions, i.e., seven for physical and five for psychological or sexual. For each question, five answers were possible: always (1), frequently (2), sometimes (3) only once (4), or never (5). For analysis purposes, exposure to physical violence was considered when participants answered 1, 2 or 3 in at least four questions, while for psychological exposure participants should have answered 1, 2 or 3 in at least three questions and for sexual violence 1, 2, 3 or 4 in at least three questions. Similarly, current social support, access to health services and trans pride were also based on specific questions, i.e., 20, 14, and 8 questions, respectively. For social support, possible answers were never (1), sometimes (2), frequently (3), very frequently (4), or always (5). Lack of current social support was considered when participant answered 1 or 2 in at least eleven questions. For access to health services possible answers were always (1), frequently (2), sometimes (3), rarely (4), or never (5). Insufficient access to health services was considered when participants answered 3, 4 or 5 in at least eight questions. Finally, answers related to trans pride questions were totally disagree (1), frequently disagree (2), does not know (3), frequently agree (4), or totally agree (5). Lack of trans pride was considered when participants answered 1, 2 or 3 in at least five questions.

## Data analysis

Data analysis took into consideration the complex sampling design of the recruitment by RDS methodology, i.e. the dependence between observations resulting from referral chains, and the probabilities of unequal selections due to the different sizes of each participant's network [32, 33]. Each one of the three cities was considered as a stratum. In each stratum, the weigh was inversely proportional to the size of each participant's network, totaling the stratum size (RDS-II estimator). The questions in the questionnaire that measured the network size of each TGW were: "How many *travestis*/trans women do you know, by name/nickname and who also know you by your name/ nickname, who live, work or study in your city?" Out of those you mentioned, how many have you met or spoken to personally, by phone or Facebook/ WhatsApp within the last 30 days?" A descriptive analysis of the weighted prevalence of SMDD with, 95% confidence intervals (95%CI) was conducted. The magnitude of the associations between the study variables and SMDD was accessed by the weighted odds ratios (wOR) with 95%CI in the bivariate and multivariate analyses. For this analysis, those with mild/moderate or moderately severe/severe SMDD were, each, compared to those without symptoms using complex multinomial logistic regression. The variables with p-value $\leq 0.20$ in the bivariate analysis were selected to start modeling and only those with p-value $< 0.05$ remained in the final model, using backwards stepwise procedure. Hosmer-Lemeshow test was used to assess fitness of the final model [34]. The analysis was conducted using the library for complex samples of STATA software version 14 (StataCorp, 2015) [35]. The study protocol was submitted for review and approved by the Sergio Arouca National School of Public Health (ENSP/ FIOCRUZ) IRB (CAAE-49359415.9.0000.5240). Written informed consent was obtained from all participants, who could withdraw consent at any stage of the process or skip any questions perceived as too sensitive, personal or distressing.

## Results and discussion

Among 864 recruited TGW, 17 seeds were excluded, leaving 847 TGW available for analysis (n = 847). The overall prevalence of SMDD was 70.1% (n = 594; 95%CI = 65.8–74 .0), whereas

51.1% (n = 418; 95% CI = 46.6–55.6) were classified as mild/moderate SMDD and 18.9% (n = 176; 95%CI = 15.8–22.6) as moderately severe/severe SMDD. The bivariate analysis indicated that history of sexual, psychological, and physical violence, history of police arrest, lack of social support and family support during childhood, and poor self-rated quality of life had statistically significant (p<0.05) higher prevalence in both groups, mild/moderate and moderately severe/severe SSMDD, as compared to those with no symptoms of depression. However, history of sex work and lack of family support during transition were only statistically associated (p<0.05) with moderately severe/severe SMDD (Table 1).

The final adjusted model indicated, that history of sexual violence (OR = 2.06, 95%CI: 1.15–3.68), history of physical violence (OR = 2.09, 95%CI: 1.20–3.67),) and poor self-rated quality of life (OR = 2.14, 95%CI: 1.31–3.49) were associated with mild/moderate SMDD. Similarly, history of sexual violence (OR = 3.02, 95%CI: 1.17–7.77), history of physical violence (OR = 4.3.62, 95%CI: 1.88–6.96), poor self-rated quality of life (OR = 3.32, 95%CI: 1.80–6.12), were also associated with moderately severe/severe SMDD. However, it should be noted that the magnitude of the ORs among those with moderately severe/severe SMDD were higher than among those with mild/moderate SMDD. In addition, lack of social current support (OR = 2.53, 95%IC: 1.31–4.88) and lack of family support in childhood (OR = 2.17, 95%IC 1.16–4.05) were only statistically associated (p<0.05) for those with more severe SMDD (Table 2). This may indicate that, although the overall prevalence of more severe SMDD is lower, higher and additional exposure to these indicators increase the likelihood of moderately severe/severe SMDD among TGW in Northeast Brazil. (Table 2).

The present study corroborates findings from other studies that reported much higher prevalence of SMDD as compared to the general population [24, 20, 36, 37], but never before in this given context. Lerri et al, (2017), despite analyzing a small sample of TGW in Brazil, found a prevalence of 80.5% of MDD [28]. In a study conducted with TGW in Canada, also using RDS, 61.2% of the sample met criteria for MDD [38]. Budge & Howard (2013), in a US study, also found a high prevalence of MDD in TGW, totaling 51.4% of participants [2]. Chodzen et al. (2019), in a New Zealand study of young transgender and youth with non-conforming gender identity, found a prevalence of MDD of 33% [39]. Despite considerable differences in estimates, and methodology used, especially depression criteria, all studies indicated considerably higher rates in TGW than in the general population.

Sexual violence has been a key predictor variable of mild/moderate SMDD in our as well as international studies. Sexual violence, especially during childhood, is a well-known risk factor for MDD [40, 41]. Among TGW, this situation is not different, with studies identifying sexual violence as a risk factor for MDD [42, 43]. Nemoto et al. (2011) pointed out that TGW are more vulnerable to sexual abuse once engaged in commercial sex work, which also emerges as a risk factor for MDD in both TGW and the general population [14, 44, 45]. Most TGW in this study reported to have engaged in commercial sex work.

Poor self-rated quality of life was statistically associated with SMDD in our study, with a stronger effect among those with moderate severe/severe depression symptoms. Studies on the association between quality of life and MDD found significant associations, regardless of whether MDD was considered as a dependent or independent variable, which suggests that positive self-perception of quality of life is protective for MDD, and MDD is a risk predictor for the worsening of quality of life, indicating potential reverse causality typical of cross-sectional studies [46–51].

Our study suggested the role of family support in childhood as key to prevent severe SMDD among TGW over time, but studies with this population are rare, especially in Brazil. Baptista et al (2001) highlighted the relationship between the lack of family support in childhood and MDD in adolescence in the general population [52]. Seibel et al (2018) found that family

**Table 1. Descriptive and bivariate analyses of symptoms of SMDD according to study variables, among transgender women in Northeast Brazil, 2017.**

| Variables | N[1] | No Symptoms %[2] | Mild/moderate symptoms of SMDD | | | | Moderately severe/severe symptoms of SMDD | | | |
|---|---|---|---|---|---|---|---|---|---|---|
| | | | %[2] | OR[3] | CI 95% | p-value | %[2] | OR[4] | 95%CI% | p-value |
| *Socio-demographic* | | | | | | | | | | |
| **Age** | | | | | | | | | | |
| >30 | 264 | 32.3 | 48.7 | 1.00 | | 0.421 | 19.1 | 1.00 | | |
| < = 30 | 583 | 28.8 | 52.3 | 1.20 | 0.77–1.89 | | 18.9 | 1.00 | 0.64–1.91 | 0.717 |
| **Schooling** | | | | | | | | | | |
| Incomplete high school or more | 369 | 31.9 | 48.7 | 1.00 | | 0.255 | 19.4 | 1.00 | | |
| Complete elementary school | 465 | 28.0 | 54.6 | 1.28 | 0.84–1.94 | | 17.4 | 1.02 | 0.60–1.73 | 0.940 |
| **Monthly Income** | | | | | | | | | | |
| >R$ 1000.00 | 230 | 29.9 | 49.6 | 1.04 | 0.64–1.69 | 0.874 | 20.6 | 1.00 | | |
| < = R$ 1000.00 | 617 | 30.0 | 51.7 | | | | 18.3 | 0.89 | 0.49–1.62 | 0.701 |
| **Race/skin color** | | | | | | | | | | |
| Non Black | 134 | 31.0 | 48.4 | 1.00 | | 0.600 | 20.6 | 1.00 | | 0.906 |
| Black | 701 | 29.0 | 52.4 | 1.15 | 0.68–1.97 | | 18.6 | 0.96 | 0.51–1.81 | |
| **Religion** | | | | | | | | | | |
| Yes | 586 | 30.3 | 51.2 | 1.00 | | 0.864 | 18.4 | 1.00 | | 0.622 |
| No | 261 | 29.0 | 50.9 | 1.04 | 0.66–1.63 | | 20.2 | 1.15 | 0.66–2.00 | |
| **Sex Work over Lifetime** | | | | | | | | | | |
| No | 200 | 37.3 | 50.5 | 1.00 | | 0.124 | 12.2 | 1.00 | | 0.003 |
| Yes | 647 | 26.6 | 51.4 | 1.43 | 0.91–2.25 | | 22.0 | 2.53 | 1.36–4.69 | |
| *History of Violence* | | | | | | | | | | |
| **Lifetime Sexual Violence[5]** | | | | | | | | | | |
| No | 103 | 49.6 | 41.7 | 1.00 | | 0.004 | 8.7 | 1.00 | | 0.001 |
| Yes | 744 | 26.9 | 52.6 | 2.33 | 1.32–4.12 | | 20.5 | 4.34 | 1.80–10.50 | |
| **Lifetime Psychological Violence [5]** | | | | | | | | | | |
| No | 225 | 41.3 | 48.0 | 1.00 | | 0.014 | 10.7 | 1.00 | | 0.000 |
| Yes | 618 | 25.6 | 5.2 | 1.76 | 1.12–2.76 | | 22.2 | 3.34 | 1.73–6.43 | |
| **Lifetime Physical Violence [5]** | | | | | | | | | | |
| No | 607 | 34.3 | 50.4 | 1.00 | | 0.001 | 15.3 | 1.00 | | 0.000 |
| Yes | 226 | 15.0 | 53.5 | 2.43 | 1.45–4.07 | | 31.6 | 4.73 | 2.62–8.53 | |
| **History of police arrest** | | | | | | | | | | |
| No | 608 | 33.9 | 50.0 | 1.00 | | 0.045 | 17.1 | 1.00 | | 0.009 |
| Yes | 239 | 21.6 | 54.3 | 1.65 | 1.01–2.71 | | 24.1 | 2.15 | 1.21–3.82 | |
| **Lifetime discrimination** | | | | | | | | | | |
| No | 91 | 38.1 | 50.3 | 1.00 | | 0.364 | 11.6 | 1.00 | | 0.118 |
| Yes | 756 | 28.8 | 51.2 | 1.35 | 0.71–2.57 | | 20.0 | 2.27 | 0.81–6.35 | |
| *Social and family support* | | | | | | | | | | |
| **Current Social Support[5]** | | | | | | | | | | |
| Yes | 551 | 35.3 | 52.1 | 1.00 | | 0.013 | 12.6 | 1.00 | | 0.000 |
| No | 260 | 18.1 | 50.0 | 1.87 | 1.14–3.07 | | 32.0 | 4.95 | 2.80–8.75 | |
| **Transgender Pride[5]** | | | | | | | | | | |
| Yes | 45 | 32.1 | 43.4 | 1.00 | | 0.584 | 24.51 | 1.00 | | 0.614 |
| No | 800 | 29.2 | 52 | 1.28 | 0.53–3.13 | | 18.1 | 0.79 | 0.32–1.98 | |
| **Family Support in Childhood** | | | | | | | | | | |
| Yes | 448 | 38.9 | 48.8 | | 1.00 | 0.000 | 12.3 | 1.00 | | 0.000 |
| No | 385 | 19.3 | 54.2 | 2.23 | 1.45–3.44 | | 26.4 | 4.33 | 2.53–7.39 | |
| **Family Support during Transition** | | | | | | | | | | |

*(Continued)*

Table 1. (Continued)

| Variables | N[1] | No Symptoms %[2] | Mild/moderate symptoms of SMDD %[2] | OR[3] | CI 95% | p-value | Moderately severe/severe symptoms of SMDD %[2] | OR[4] | 95%CI% | p-value |
|---|---|---|---|---|---|---|---|---|---|---|
| Yes | 675 | 32.5 | 52.2 | 1.00 | | 0.293 | 15.4 | 1.00 | | 0.001 |
| No | 171 | 22.0 | 47.7 | 1.35 | 0.77–2.36 | | 30.3 | 2.91 | 1.56–5.40 | |
| **Positive Self Rated Life Quality** | | | | | | | | | | |
| Yes | 481 | 38.3 | 48.0 | 1.00 | | 0.000 | 13.6 | 1.00 | | 0.000 |
| No | 360 | 18.5 | 55.7 | 2.40 | 1.53–3.76 | | 25.8 | 3.92 | 2.26–6.78 | |
| **Living Alone** | | | | | | | | | | |
| No | 653 | 31.1 | 51.4 | 1.00 | | 0.487 | 17.5 | 1.00 | | 0.161 |
| Yes | 194 | 25.1 | 49.9 | 1.20 | 0.71–2.04 | | 25.0 | 1.78 | 0.97–3.25 | |
| *Health Related* | | | | | | | | | | |
| **Current use of hormones** | | | | | | | | | | |
| No | 357 | 28.7 | 50.6 | 1.00 | | 0.822 | 20.65 | 1.00 | | 0.542 |
| Yes | 434 | 28.9 | 53.5 | 1.05 | 0.68–1.63 | | 17.60 | 0.85 | 0.50–1.44 | |
| **Access to Health Services[5]** | | | | | | | | | | |
| Yes | 260 | 32.4 | 48.7 | 1.00 | | 0.246 | 19.0 | 1.00 | | 0.277 |
| No | 382 | 26.1 | 52.7 | 1.34 | 0.82–2.21 | | 21.2 | 1.38 | 0.77–2.49 | |
| **Living with HIV/AIDS** | | | | | | | | | | |
| No | 621 | 29.4 | 52.1 | 1.00 | | 0.496 | 18.5 | 1.00 | | 0.580 |
| Yes | 193 | 32.1 | 46.6 | 0.82 | 0.50–1.35 | | 21.3 | 1.05 | 0.58–1.91 | |
| **Use of ART** | | | | | | | | | | |
| No | 761 | 29.6 | 52.0 | 1.00 | | 0.375 | 18.4 | 1.00 | | 0.594 |
| Yes | 75 | 33.0 | 41.4 | 0.71 | 0.34–1.50 | | 25.7 | 1.25 | 0.55–2.88 | |

[1] Total sample for each category

[2] Proportion of SMDD in each category in comparison to the total (N)

[3] Weighted odds ratio comparing mild/moderate SMDD with no SMDD for each characteristic

4Weighted odds ratio comparing moderately severe/severe SMDD with no SMDD for each characteristic.

[5] See text for definition

support was associated with higher self-esteem and highlighted the importance of parental support to improve quality of life in transgender people [20]. Ryan et al (2010) underlines that family acceptance is associated with a positive young adult mental and physical health [53]. Hoffman (2014) indicated that social support without family support was not significantly

Table 2. Multivariate analysis of factors associated with symptoms of SMDD among transgender women in Northeast Brazil, 2017.

| Variables | Mild/moderate OR[1] | 95% CI | p-value | Moderately severe/severe OR[2] | 95% CI | p-value |
|---|---|---|---|---|---|---|
| Sexual violence over lifetime (yes vs no)[3] | 2.06 | 1.15–3.66 | 0.015 | 3.02 | 1.17–7.77 | 0.022 |
| Physical violence over lifetime (yes vs no)[3] | 2.10 | 1.20–3.77 | 0.010 | 3.62 | 1.88–7.00 | 0.000 |
| Family Support during childhood (no vs yes) | 1.54 | 0.93–2.54 | 0.091 | 2.17 | 1.16–4.05 | 0.015 |
| Current social support[3] | 1.26 | 0.73–2.20 | 0.408 | 2.53 | 1.31–4.90 | 0.006 |
| Positive Self-rated Quality of life (no vs yes) | 2.14 | 1.31–3.49 | 0.002 | 3.32 | 1.80–6.12 | 0.000 |

Goodness-of-Fit (Hosmer-Lemeshow Test) of the multivariate final model = 12.6; p = 0.126; 8 df)

[1] Weighted odds ratio comparing mild/moderate SMDD with no SMDD for each characteristic

[2] Weighted odds ratio comparing moderately severe/severe SMDD with no MDD for each characteristic.

[3] See text for definition.

associated with MDD, and that the association between family support and MDD may vary with age and be more relevant up to middle age [20]. Other studies also point that people who have social support have a lower prevalence of MDD compared with members of the same population in which such social support is minimum. This situation is also true in specific studies with TGW populations [19, 28, 50]. Nemoto et al. (2011) suggested that the perception of social support might be a more important protective factor than social support itself [14]. A study conducted with a transgender population in the United States pointed out that support by other transgender people can mitigate the effect of other factors on MDD in this population [15]. Support provided by social networks was found to affect positively transgender adolescents who would not otherwise get any other social support [54].

Physical violence is very frequent within the LGBTQIA+ community, especially among TGW and has an important role in the occurrence of MDD in the general population, and, among TGW [55, 56]. Our study shows an important association between physical violence and SMDD. Parente et al (2020) in a study conducted in a Northeastern town in Brazil found that most of the aggressors of LGBTQIA+ were unknown by standers [57]. Pinto et al (2020) pointed out that physical violence is the most prevalent violence in the LGBTQIA+ community, and highlight the need of compulsory notifications of these incidents in Brazil [58]. Nevertheless, studies analyzing the association between physical violence and MDD or SMDD among TGW in Brazil are still scarce.

Potential limitations of our study include the cross-sectional design, a possible dependency of the data due to RDS recruitment and a lack of information on access to MDD treatment or follow-up. Finally, these data are pre COVID-19 pandemic, and the pandemic has seen a staggering increase in the prevalence of mental health disorders worldwide, including depression [59] and this may indicate that the already high prevalence found in our study among TGW, currently, could be even higher.

## Conclusions

In the present work, we carried out an exploratory analysis of the TGW population in Northeast Brazil. The data suggest that social determinants and context variables are the main drive that affects the prevalence of SMDD in this population, especially among those with moderate severe/severe symptoms. Actions that target these determinants should be considered in prevention public policies. Further studies are still needed in order to analyze the effect either of the variables herein presented on the causal pathway of SMDD among TGW, independently or as syndemic factors. Considering that health problems are often analyzed from the cisgender population point of view in Brazil, additional studies are also necessary in order to better understand the specific needs of TGW populations, thus tailoring mental health policies towards these specific needs. Finally, it is vital not to pathologize dissident gender performances and psychological and social distresses themselves, considering these issues are influenced by a complex set of factors, including social, cultural and symbolic, among others.

## Supporting information

**S1 File.**
(XLS)

## Acknowledgments

The authors would like to express their gratitude to the participants of the study, to the local teams that carried out the fieldwork in the three cities, and all collaborating NGOs. We are

also grateful for the support of STI/HIV/AIDS and Viral Hepatitis Department of the Brazilian Minister of Health.

## Author Contributions

**Conceptualization:** Marcelo Machado de Almeida, Luís Augusto Vasconcelos da Silva, Francisco Inácio Bastos, Inês Dourado.

**Data curation:** Marcelo Machado de Almeida, Mark Drew Crosland Guimarães, Inês Dourado.

**Formal analysis:** Marcelo Machado de Almeida, Francisco Inácio Bastos, Inês Dourado.

**Funding acquisition:** Francisco Inácio Bastos, Inês Dourado.

**Investigation:** Marcelo Machado de Almeida, Inês Dourado.

**Methodology:** Marcelo Machado de Almeida, Luís Augusto Vasconcelos da Silva, Inês Dourado.

**Project administration:** Inês Dourado.

**Resources:** Inês Dourado.

**Software:** Marcelo Machado de Almeida.

**Supervision:** Marcelo Machado de Almeida, Luís Augusto Vasconcelos da Silva, Mark Drew Crosland Guimarães, Inês Dourado.

**Validation:** Luís Augusto Vasconcelos da Silva, Francisco Inácio Bastos, Mark Drew Crosland Guimarães, Carolina Coutinho, Ana Maria de Brito, Socorro Cavalcante, Inês Dourado.

**Visualization:** Luís Augusto Vasconcelos da Silva, Francisco Inácio Bastos, Mark Drew Crosland Guimarães, Carolina Coutinho, Ana Maria de Brito, Socorro Cavalcante, Inês Dourado.

**Writing – original draft:** Marcelo Machado de Almeida, Inês Dourado.

**Writing – review & editing:** Marcelo Machado de Almeida, Luís Augusto Vasconcelos da Silva, Francisco Inácio Bastos, Mark Drew Crosland Guimarães, Carolina Coutinho, Ana Maria de Brito, Socorro Cavalcante, Inês Dourado.

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
