## [Decision Letter · Decision Letter 0]

18 Nov 2021

PONE-D-21-16373Major Depression Disorder in a large sample of transgender women in Brazil: the role of protective factors in the reduction of mental health disordersPLOS ONE

Dear Dr. Almeida,

Thank you for submitting your manuscript to PLOS ONE. After careful consideration, we feel that it has merit but does not fully meet PLOS ONE’s publication criteria as it currently stands. Therefore, we invite you to submit a revised version of the manuscript that addresses the points raised during the review process.

As the reviewers commented, there are several points that should be improved. (e.g., The Introduction section needs to be more elaborated and the methodology should be clarified.)==============================

We look forward to receiving your revised manuscript.

Kind regards,

Kyoung-Sae Na, M.D.

Academic Editor

PLOS ONE

Journal Requirements:

“The authors would like to express their gratitude to the participants of the study, to the local teams that carried out the fieldwork in the three cities, and all collaborating NGOs. We are also grateful for the support of STI/HIV/AIDS and Viral Hepatitis Department of the Brazilian Minister of Health. “

“Funding: Funding for this study was provided by The Brazilian Ministry of Health through its Secretariat for Health Surveillance and its Department of Prevention, Surveillance and Control of Sexually Transmitted Infections, HIV/AIDS and Viral Hepatitis.

No funders played any role in: study design, data collection and analysis, decision to publish or preparation of the manuscript.”

Reviewers' comments:

Reviewer's Responses to Questions

**Comments to the Author**

1. Is the manuscript technically sound, and do the data support the conclusions?

Reviewer #1: Yes

Reviewer #2: Partly

2. Has the statistical analysis been performed appropriately and rigorously? 

Reviewer #1: Yes

Reviewer #2: Yes

3. Have the authors made all data underlying the findings in their manuscript fully available?

Reviewer #1: Yes

Reviewer #2: Yes

4. Is the manuscript presented in an intelligible fashion and written in standard English?

Reviewer #1: Yes

Reviewer #2: Yes

5. Review Comments to the Author

Reviewer #1: 1. In the discussion part "The present study corroborates findings from other studies that reported much higherprevalence of MDD in TGW vis-à-vis the general population. " What is vis-à-vis?

2．How sample size was set ? Is 847 enough ? The author should describe in detail.

3．Research paid to the participants. Is it possible to lead to choice bias? How did author avoid related bias ?

4．Authors used some self difined question. I’m confused that why not use some stardized scales such as social support sclale, life event scale...?

5．Have authors validated the reliability of regression model? If they have, how did the do?

Reviewer #2: This study presents findings regarding depression and correlates of depression in a sample of trangender women in Brazil. The findings in this paper present new information on correlates of depression among this understudied population.

At this point, the introduction is brief and does not introduce or justify the variables used as correlates. This introduction needs significant additions in order to justify the use of the variables under study.

The conclusion of the paper would also benefit from more development, as there is little discussion of implications of these findings for the future research that the authors state should be done.

Minor changes to make:

1. Abstract: "Transgender women (TGW) are one of the most vulnerable groups." Clarify this - most vulnerable in terms of what factors and/or compared to what groups?

2. Abstract: "Major Depression Disorder (MDD) is one of the most important mental health conditions due to an increasing trend in prevalence." Clarify that this is in the general population.

3. Introduction: "Among them, transgender women (TGW) are one of the most vulnerable groups." Are you saying that TGW are more vulnerable than others with "non-conforming gender performances?" If so, please provide a citation indicating as such. If not, clarify.

4. Methods: clarify that the MDD cut points are validated by other studies as useful for differentiating mild/moderate and severe categories.

5. Results: Line 134: Authors state all p values are below 0.20 but they're all below 0.05, which would be a stronger statement.

6. Results: p values are listed as 0.000 in several places. I would write these as p <0.0001, but check with PLOS One formatting.

7: Results on multivariate analyses: Organize findings by covariates and whether each is significant for mild/mod or severe.

8. Results, Table 2: Indicate which ORs are significant with a superscript symbol.

9. Discussion, page 14, first line. Change "statistical" to "statistically"

9. Discussion, page 14. There are 2 variables (gender discrimination, imprisonment) in which this study differed from previous findings. State more about why these might be different from previous studies.

6. PLOS authors have the option to publish the peer review history of their article (what does this mean?). If published, this will include your full peer review and any attached files.

Reviewer #1: No

Reviewer #2: **Yes: **Beth R Hoffman

---

## [Author Response · Author response to Decision Letter 0]

6 Feb 2022

REVIEWER 1 COMMENTS:

(1) In the discussion part "The present study corroborates findings from other studies that reported much higher prevalence of MDD in TGW vis-à-vis the general population." What is vis-à-vis?

R: What is vis-à-vis means “in relation to”; “with regard to”. We have changed the text accordingly (line 192). 

(2) How sample size was set? Is 847 enough? The author should describe in detail.

R: First, we would like to mention that the sample size for this study was set a priori by the Funding Agency (Department of Chronic Diseases and Sexually Transmitted Infections- DCCI- Brazilian Ministry of Health). In addition to the Transgender Women (TGW) study, two other RDS multicenter studies were conducted in Brazil (MSM and Female Sex Workers - FSW) using the same methodology and in the same cities, allowing for proper comparisons. RDS is recommended for hard-to-reach populations, such as TGW, MSM and FSW, which are difficult or even impossible to be sampled using standard probability sampling. 

Second, these studies had a primary objective of estimating the prevalence of HIV, HBV, HCV and syphilis, which are relatively rare events and differ across the cities (as compared to depression and other secondary outcomes). 

Third, as with other non-probability sample studies, RDS sample size estimation is based on the desired design effect (DE), and this is usually calculated only post hoc (Wejnert et al,2012) https://www.ncbi.nlm.nih.gov/pmc/articles/PMC3382647/

 Fourth, however, because these were multicity studies, and the prevalence of HIV varies, the Funders established between 250 and 350 participants per city, and this was based on a priori DE of 2. Therefore, the sample size used in our study was clearly and with enough power to estimate the prevalence of depression and its associated factors, which was much higher than the prevalence of HIV, used by the Funders to estimated DE and the desired sample size. 

In conclusion, although we understand it is beyond the scope of this manuscript to discuss this issue in depth, we agree with the reviewer that the manuscript should at least briefly indicate how sample size was estimated. This way, we have changed the text (lines 141-50), in order to clarify this point. 

Besides empirical studies, members of our research team (FIB and coworkers) have implemented simulations studies (Sperandei et al., 2018. Sperandei et al., in press) and such platform has been used to simulate RDS studies using different families of random graph models: Erdös-Renyi; Watts-Strogatz; Barabasi-Albert & Interconnected Islands.

For all underlying graph models, it´s possible to analyze events of interest (i.e. those which are more prevalent and have a key role in conceptual models) with the necessary precision (Rothman & Greenland, 2018).

Of course, the proper analysis of less prevalent outcomes and covariates is precluded by the small sample size and the limitations which are intrinsic to non-probability samples (whatever the method one might choose; Elliot and Valliant, 2017) 

They constitute an insurmountable limitation of this study and the vast majority of studies using RDS, worldwide. 

References for the cited simulation studies:

https://www.sciencedirect.com/science/article/abs/pii/S0378873316301769

Sperandei, S; Bastos, LS; Ribeiro-Alves, M; Reis, A; Bastos, FI, Assessing Logistic Regression Applied to Respondent-Driven Sampling Studies: A simulation study with an application to empirical data. International Journal of Social Research Methodology (in press)

Rothman and Greenland: https://pubmed.ncbi.nlm.nih.gov/29912015/

(3) Research paid to the participants. Is it possible to lead to choice bias? How did author avoid related bias?

R: Yes, both, recruitees and recruiters received paid incentives, similar to the other RDS studies mentioned above (MSM and FSW). We should note that the project was approved by local and national ethical review boards in all aspects, including incentives. This double incentive system is an intrinsic part of the RDS method since the first publications of its originator (D. Heckathorn). 

Notwithstanding the debates respecting the possibility of biasing findings as well as respecting the ethical aspects of such incentives since the inception of the first RDS-studies (e.g. https://pubmed.ncbi.nlm.nih.gov/20167881/), the overall conclusion is that such incentives are acceptable from an ethical point of view and that biases can be minimized (but not averted) by procedures usually adopted, such as the use of careful designed estimators and the systematic use of weighting methods. As indicated in the method section, our estimators were properly weighted by the inverse probability of the self-reported network size. Although there is no clear way to make any form of inference based on non-probability samples as solid as inference anchored in probability samples, it is not possible to use probability sampling for hard-to-reach populations, as mentioned above. These are difficulties which cannot be eliminated in the context of RDS studies (https://pubmed.ncbi.nlm.nih.gov/20351258/,) but science moves ahead despite such caveats. 

Despite its incredible accuracy respecting the most different calculations, a hundred years after its first formulations, Quantum Mechanics remains affected by key unexplained issues and remains fully incompatible with General Relativity at the micro-level (see, for instance: Smolin, 2020) https://www.amazon.co.uk/Einsteins-Unfinished-Revolution-Search-Quantum/dp/014197916X/ref=sr_1_3?crid=82N90QSH8XQI&keywords=lee+smolin+books&qid=1642430186&s=books&sprefix=lee+smolin%2Cstripbooks%2C268&sr=1-3

(4) Authors used some self-defined question. I’m confused that why not use some standardized scales such as social support scale, life event scale...?

R: As indicated above, this study was funded by the Department of Chronic Diseases and Sexually Transmitted Infections- DCCI- Brazilian Ministry of Health along with two other RDS (MSM and FSW). For the purpose of generating common and standardized questions of interest for all three studies (e.g use of health services, previous STI, HIV testing) the questionnaires were designed and tested by the DCCI. For each project, a team of experts assessed the questionnaires and suggested additional specific topics and questions pertinent to each of the populations studied (e.g. hormone use in the case of TGW). We should note that the DCCI has conducted previous RDS studies when most standard questions were applied. In addition, we should also note that the PHQ-9 scale used to define our outcome of interest was standard for all three projects, and has been extensively validated in Brazil.

(5) Have authors validated the reliability of regression model? If they have, how did they do?

R: Indeed, Goodness-of-Fit (GOF) of the final model was assessed by Hosmer-Lemeshow test. As indicated, the final model was considered adequate (Chi square=12.6, 8 DF, p=0.126). We have therefore clarified the text in the method and result sections. 

REVIEWER 2 COMMENTS:

This study presents findings regarding depression and correlates of depression in a sample of trangender women in Brazil. The findings in this paper present new information on correlates of depression among this understudied population.

At this point, the introduction is brief and does not introduce or justify the variables used as correlates. This introduction needs significant additions in order to justify the use of the variables under study.

The conclusion of the paper would also benefit from more development, as there is little discussion of implications of these findings for the future research that the author’s state should be done.

R: Thank you for this comment. We have revised and adjusted the text accordingly.

MINOR CHANGES TO MAKE

(1) Abstract: "Transgender women (TGW) are one of the most vulnerable groups." Clarify this - most vulnerable in terms of what factors and/or compared to what groups?

R: Thank you for this comment. It is known that TGW are more vulnerable to a myriad of health problems (and more specifically to mental health problems) when compared to LGBTQA+ people, cisgender women or general population. (Dhejne C, Van Vlerken R, Heylens G, Arcelus J. Mental health and gender dysphoria: A review of the literature. Int Rev Psychiatry. 2016;28(1):44-57. doi: 10.3109/09540261.2015.1115753. PMID: 26835611.)

We have therefore changed the text in order to clarify this topic (lines 3 and 4). 

(2) Abstract: "Major Depression Disorder (MDD) is one of the most important mental health conditions due to an increasing trend in prevalence." Clarify that this is in the general population.

R: Thank you for this comment. We have revised and adjusted the text accordingly (line 5).

(3) Introduction: "Among them, transgender women (TGW) are one of the most vulnerable groups." Are you saying that TGW are more vulnerable than others with "non-conforming gender performances?" If so, please provide a citation indicating as such. If not, clarify.

R: Thank you for this comment. We have revised and adjusted the text accordingly. 

“Among them, transgender women (TGW) are one of the most vulnerable groups. Specifically, mental disorders such as anxiety and mood disorders are more prevalent in TGW than in the general population”

(4) Methods: clarify that the MDD cut points are validated by other studies as useful for differentiating mild/moderate and severe categories.

R: Yes. PHQ-9 is validated in Brazil with the same original cut points. (Santos I S., Tavares BF, Munhoz TN., Almeida LSPD, Silva NTBD, Tams BD, et al . Sensibilidade e especificidade do Patient Health Questionnaire-9 (PHQ-9) entre adultos da população geral. Cad. Saúde Pública [Internet]. 2013 Aug [cited 2020 Sep 12] ; 29( 8 ): 1533-1543. Available from: http://www.scielo.br/scielo.php?script=sci_arttext&pid=S0102-311X2013000800006&lng=en. http://dx.doi.org/10.1590/0102-311X00144612)

(5) Results: Line 134: Authors state all p values are below 0.20 but they're all below 0.05, which would be a stronger statement.

R: Thank you for this comment. We have revised and adjusted the text accordingly (lines 146-48)> All variables with p < 0.20 in the bivariate analysis. We have clarified our analysis in the manuscript indicating that variables which were significant at p < 0.20 in the bivariate analysis were included to start modeling, and only those significant at p < 0.05 remained in the final model. Therefore, we should note that Table 2 now only includes the final model result, upon which we ran GOF assessment as explained to Reviewer one, comment five above. 

(6) Results: p values are listed as 0.000 in several places. I would write these as p <0.0001, but check with PLOS One formatting.

R: Thank you for this comment. We have revised and adjusted the text accordingly.

(7) Results on multivariate analyses: Organize findings by covariates and whether each is significant for mild/mod or severe.

R: Thank you for this comment. We have revised and adjusted the text accordingly.

(8) Results, Table 2: Indicate which ORs are significant with a superscript symbol.

R: Thank you for this comment. We have revised and adjusted the text accordingly.

(9) Discussion, page 14, first line. Change "statistical" to "statistically"

R: Thank you for this comment. We have revised and adjusted the text accordingly.

(10) Discussion, page 14. There are 2 variables (gender discrimination, imprisonment) in which this study differed from previous findings. State more about why these might be different from previous studies.

R: Thank you for this comment. The variables gender discrimination or imprisonment did not remain in the final model, although in the bivariate analysis there was a positive association between these variables and depression (p value= 0.118 and 0.009, respectively) which is corroborated by the literature. However, because of the extent of the discussion we have decided to focus only on the variables of the final model with p< 0.05, indicating whether they corroborate or contradicts other publications worldwide. This way, we decided to remove from the discussion those variables that had no statistical significance at the p< 0.05 level in the multivariate analysis.

---

## [Decision Letter · Decision Letter 1]

18 Apr 2022

Factors associated with symptoms of major depression disorder among transgender women in Northeast Brazil

PONE-D-21-16373R1

Dear Dr. Almeida,

We’re pleased to inform you that your manuscript has been judged scientifically suitable for publication and will be formally accepted for publication once it meets all outstanding technical requirements.

Kind regards,

Kyoung-Sae Na, M.D., Ph.D.

Academic Editor

PLOS ONE

Additional Editor Comments (optional):

Reviewers' comments:

Reviewer's Responses to Questions

**Comments to the Author**

1. If the authors have adequately addressed your comments raised in a previous round of review and you feel that this manuscript is now acceptable for publication, you may indicate that here to bypass the “Comments to the Author” section, enter your conflict of interest statement in the “Confidential to Editor” section, and submit your "Accept" recommendation.

Reviewer #2: All comments have been addressed

2. Is the manuscript technically sound, and do the data support the conclusions?

Reviewer #2: Yes

3. Has the statistical analysis been performed appropriately and rigorously? 

Reviewer #2: Yes

4. Have the authors made all data underlying the findings in their manuscript fully available?

Reviewer #2: Yes

5. Is the manuscript presented in an intelligible fashion and written in standard English?

Reviewer #2: Yes

6. Review Comments to the Author

Reviewer #2: (No Response)

7. PLOS authors have the option to publish the peer review history of their article (what does this mean?). If published, this will include your full peer review and any attached files.

Reviewer #2: **Yes: **Beth R Hoffman

---

## [Editor Report · Acceptance letter]

16 Aug 2022

PONE-D-21-16373R1 

Factors associated with symptoms of major depression disorder among transgender women in Northeast Brazil 

Dear Dr. Almeida:

I'm pleased to inform you that your manuscript has been deemed suitable for publication in PLOS ONE. Congratulations! Your manuscript is now with our production department. 

Kind regards, 

on behalf of

Dr. Kyoung-Sae Na 

Academic Editor

PLOS ONE